# A Multi-mRNA Host-Response Molecular Blood Test for the Diagnosis and Prognosis of Acute Infections and Sepsis: Proceedings from a Clinical Advisory Panel

**DOI:** 10.3390/jpm10040266

**Published:** 2020-12-07

**Authors:** James Ducharme, Wesley H. Self, Tiffany M. Osborn, Nathan A. Ledeboer, Jonathan Romanowsky, Timothy E. Sweeney, Oliver Liesenfeld, Richard E. Rothman

**Affiliations:** 1Department of Medicine, McMaster University, Hamilton, ON L8S 4L8, Canada; paindoc22000@yahoo.com; 2Department of Emergency Medicine, Vanderbilt University Medical Center, Nashville, TN 37220, USA; wesley.self@vumc.org; 3Department of Medicine, Division of Emergency Medicine and Department of Surgery, Washington University, St. Louis, MO 63110, USA; osbornt@wustl.edu; 4Department of Pathology, Medical College of Wisconsin, Milwaukee, WI 53226, USA; nledeboe@mcw.edu; 5Inflammatix Inc., Burlingame, CA 94010, USA; jonathan@inflammatix.com (J.R.); tim@inflammatix.com (T.E.S.); 6Department of Emergency Medicine, The Johns Hopkins University, Baltimore, MD 21264, USA; rrothma1@jhmi.edu

**Keywords:** acute infections, sepsis, bacterial, viral, diagnosis, prognosis, point-of-care, emergency medicine

## Abstract

Current diagnostics are insufficient for diagnosis and prognosis of acute infections and sepsis. Clinical decisions including prescription and timing of antibiotics, ordering of additional diagnostics and level-of-care decisions rely on understanding etiology and implications of a clinical presentation. Host mRNA signatures can differentiate infectious from noninfectious etiologies, bacterial from viral infections, and predict 30-day mortality. The 29-host-mRNA blood-based InSep^TM^ test (Inflammatix, Burlingame, CA, formerly known as HostDx^TM^ Sepsis) combines machine learning algorithms with a rapid point-of-care platform with less than 30 min turnaround time to enable rapid diagnosis of acute infections and sepsis, as well as prediction of disease severity. A scientific advisory panel including emergency medicine, infectious disease, intensive care and clinical pathology physicians discussed technical and clinical requirements in preparation of successful introduction of InSep into the market. Topics included intended use; patient populations of greatest need; patient journey and sample flow in the emergency department (ED) and beyond; clinical and biomarker-based decision algorithms; performance characteristics for clinical utility; assay and instrument requirements; and result readouts. The panel identified clear demand for a solution like InSep, requirements regarding test performance and interpretability, and a need for focused medical education due to the innovative but complex nature of the result readout. Innovative diagnostic solutions such as the InSep test could improve management of patients with suspected acute infections and sepsis in the ED, thereby lessening the overall burden of these conditions on patients and the healthcare system.

## 1. Introduction

Sepsis is a leading cause of morbidity, mortality, and health system costs. It is a major driver impacting health system costs across the United States (US) where federal Centers for Medicare and Medicaid (CMS) payments are partially tied to sepsis bundle performance criteria [1,2,3,4]. Sepsis is a syndrome most recently redefined (‘Sepsis-3’) as a life-threatening dysregulated host immune response to infection [1]. Although frequently used synonymously with bacteremia, sepsis can be caused by bacterial as well as viral, fungal, or parasitic infections. Sepsis can also be mimicked by noninfectious etiologies, making diagnosis particularly challenging where an infectious source is not immediately apparent [5].

Early treatment with appropriate antibiotics has been shown to reduce morbidity and mortality in cases of bacterial sepsis [6,7]. However, antibiotics are well known to have substantial adverse side effects. One in five hospital patients treated with parenteral antibiotics experiences an adverse drug event, including *Clostridioides difficile* infection [8,9]; further, indiscriminate and prolonged use of antibiotics can lead to antimicrobial resistance [10]. Thus, rapid and accurate diagnosis of sepsis, and its underlying cause, is of great interest [11,12].

From a clinical perspective, sepsis can be visualized along two “axes”, namely, the presence of an infection and the severity of the condition (Figure 1A) [12,13]. Making the diagnosis of infection and predicting prognosis of illness remains challenging with each leading to separate, but intertwined, clinical actions (Figure 1B). The diagnostic dilemma is further complicated by the time required to obtain diagnostic results. Especially during the proximal phase of presentation, clinicians may need to make treatment decisions with incomplete and/or imperfect data.

Current diagnostic tools for sepsis include physical exam findings and vital signs, which are often nonspecific, as well as a large variety of clinical laboratory, imaging, and microbiological findings [12]. While many diagnostic tests add information to the overall clinical picture, they may not provide accurate information regarding the expected clinical trajectory or may not be clinically actionable at the point initial treatment decisions are required. Up to 50% of patients who are admitted through the emergency department (ED) and go on to severe sepsis or septic shock in the first 48 h of admission are not identified in the ED pointing towards the need for testing solutions that can identify those at risk for deterioration prior to physiologic decline. The lack of actionable results provided by most blood culture collections is well documented, as is the fact that the majority of patients presenting with acute infection to the ED have localized infections, which are not detectable by blood culture or DNA-based pathogen detection tests [14,15,16]. As many patients suspected of sepsis in acute care settings have localized infections, and that the current recommendations under the Surviving Sepsis Campaign guidelines are to treat these patients as soon as possible after recognition and within one hour for both sepsis and septic shock [17], the majority of these patients receive empiric antibiotic therapy [5,18,19]. However, broad antibiotic therapy is associated with adverse side effects in up to 1 in 5 patients, as well as contributing significantly to antimicrobial resistance [11]. This underscores the need for a rapid, accurate diagnostic test solution that can be administered at the point of care.

The Surviving Sepsis Campaign guidelines and the CMS “SEP-1” bundle are characterized by (1) source control and antimicrobial therapy to fight a presumed underlying infection, and (2) supportive care to maintain physiology [20,21]. Most diagnostics currently used for the diagnosis of sepsis result in clinical actions on source control and treatment with antibiotics. Roughly two-thirds of all septic patients enter the hospital through the ED [22]. Given that these septic patients must be treated with antibiotics within 60 min to obtain optimal outcomes [7,23,24], only those diagnostic tests with a turnaround time substantially shorter than the 60 min “window” will be most useful in the ED workflow.

While the early administration of antimicrobials is critical in patients at risk for sepsis, as described above, unnecessary (over) use has been identified as a major clinical and public health issue [11]. Many agencies including the Centers for Disease Control and Prevention (CDC), Biomedical Advanced Research and Development Authority (BARDA), World Health Organization (WHO) and professional societies such as the Infectious Disease Society of America (IDSA) have focused on advancing strategies to prevent the overuse of antibiotics in acute care via antimicrobial stewardship [25]. Studies indicate that 30–50% of antibiotics prescribed in hospitals are unnecessary or inappropriate [26]. Misuse occurs in healthcare settings for a variety of reasons, including use of antibiotics when not needed, continued treatment when no longer necessary, inappropriate dosing, use of broad-spectrum agents instead of specific directed ones to treat specific highly susceptible bacteria, and wrong antibiotic to treat an infection [27]. In addition to antimicrobial stewardship, diagnostic stewardship, defined as the process to select the right test for the right patient generating accurate, clinically relevant results at the right time to optimally influence clinical care and to conserve health care resources has been identified as a key to successful reduction of antibiotic use [28,29].

An ideal diagnostic test for the management of patients with acute infections and suspected sepsis would answer the following clinically actionable questions in a highly accurate and rapid (<30 min) manner: (1) does the patient require antibiotics?, (2) what other diagnostic tests should be ordered to make the diagnosis?, and (3) what level of care does the patient require? Development of such a test would facilitate to address several goals to be realized for patients in the ED, not limited to:i.avoidance of antibiotics in patients without bacterial infections (i.e., “rule-out” bacterial infection),ii.early identification of bacterial infections in patients clinically suspected of viral or noninfectious inflammatory conditions (i.e., “rule-in” bacterial infection),iii.determining the optimal level of care (e.g., observation instead of admission or discharge instead of observation) for a patient with lower severity (de-escalate).iv.predicting organ dysfunction and decompensation over a short period to decide on closer monitoring and possibly intensive care unit (ICU) transfer (escalate).

Inflammatix is developing an innovative molecular host response test, InSep, designed to meet the needs outlined above. The test is designed to quantitate 29-host-mRNAs from whole blood in less than 30 min and will report three scores for each test ordered: (1) likelihood of a bacterial infection; (2) likelihood of a viral infection; and (3) severity of the condition (as defined below). In January 2019, a clinical advisory panel was convened to gather expert clinician and laboratorian feedback on the target product profile. Here we outline the major themes identified in that meeting.

## 2. Methods

To gain further understanding on key assay and instrument characteristics, clinical and laboratory workflow and environment, as well as implementation of the 29-host-mRNA test, InSep, a one-day scientific advisory panel was held. Clinical advisors practicing in the US and Canada included two academic emergency physicians, one nonacademic emergency physician, one academic clinical laboratory specialist, one infectious disease/clinical microbiology specialist and two physicians trained in emergency medicine and critical care who work in both environments. Formal presentations and semi-structured open informed discussions were used to introduce the following topics: HostDx Sepsis Overview, Defining Intended Use, Result Readout, Special Populations, Future Indications, Patient Flows and Assay and Instrument Requirements. Transcripts were recorded and utilized in preparing this manuscript. As the test has since been rebranded from HostDx Sepsis to InSep, to avoid confusion, we have replaced references to the test with “InSep”.

### 2.1. Introduction to InSep

In order to elicit feedback on InSep, the following product profile was presented to the advisory board and discussed. Although the final InSep product has changed in some respects since the time of the presentation, the present information was provided to the advisory panel at the time.

#### 2.1.1. Assay and Instrument Overview

InSep is a 29-host-mRNA blood test that interprets patterns in the immune system to allow (1) diagnosis of acute infections; (2) differentiation of bacterial from viral infections; and (3) prognosis of disease severity from whole blood. The test uses 2.5 mL of whole blood collected in PAXgene^®^ Blood RNA tubes (Food and Drug Administration (FDA)-cleared, though not yet for use with InSep). The sample is then processed in a closed cartridge on a point-of-need platform currently in development. Custom, optimized protocols are used for sample preparation and RNA purification. No leukocyte separation is carried out; RNA is extracted direct from the sample using magnetic beads in the cartridge. Isothermal amplification is then used to amplify and quantitate the 29-mRNAs (plus housekeeping genes), which are then fed to machine learning algorithms to produce actionable scores (see Section 2.1.4). The hands-on time required for the instrument workflow is less than 2 min, with no pipetting required. Figure 2 shows the proposed workflow from sample collection to result generation.

#### 2.1.2. Identification of the 29-mRNA Classifier in InSep

Previously described are three nonoverlapping host response-based mRNA scores that could (1) diagnose the presence or absence of an acute infection (11 mRNAs) [30], (2) distinguish an infection as bacterial or viral (7 mRNAs) [31], and (3) determine the risk of 30-day mortality from sepsis (11 mRNAs) [32]. The public and private databases mined for this purpose included heterogeneous cohorts from diverse geographies, ethnicities, age groups, diagnoses, across outpatients, patients in emergency departments, inpatients on general wards, and ICUs. Each of these three previously described scores is calculated as the difference in geometric means of the expression values of two gene “modules” (one composed of overexpressed genes and the other composed of underexpressed genes). The modules are: (1) infection-up: *CEACAM1, ZDHHC19, C9orf95, GNA15, BATF, C3AR1;* (2) infection-down: *KIAA1370, TGFBI, MTCH1, RPGRIP1, HLA-DPB1*; (3) bacterial-viral-up: *HK3, TNIP1, GPAA1, CTSB*; (4) bacterial-viral-down: *IFI27, JUP, LAX1*; (5) mortality-up: *DEFA4, CD163, RGS1, PER1, HIF1A, SEPP1, C11orf74, CIT*; and (6) mortality-down: *LY86, TST, KCNJ2*. Housekeeping gene(s) act as internal controls. The above genes may change in the final product as they have not been optimized for isothermal amplification.

A “version 1” machine learning classifier algorithm that reads the expression levels of the 29-target mRNAs and generates a score for each bacterial, viral, noninfected class using neural networks has been described by Mayhew et al. [33]. A separate machine learning classifier forms the severity prediction score. The signatures described above have been validated using independent databases [34,35] as well as in an independent validation study in critically ill patients [36], where procalcitonin (PCT) was shown to have marginal performance compared to InSep. The results of these studies confirm that heterogeneous data form a suitable training dataset for these new metrics, and that tests based on transcriptomics may outperform single-protein biomarkers such as procalcitonin.

#### 2.1.3. Description of the Classifier Results

InSep will report results in three separate readouts, reflecting: (a) the likelihood of a bacterial infection, (b) the likelihood of a viral infection, and (c) the severity of the condition. Each result will contain a numerical score and will fall into one of four interpretation bands for the bacterial, viral or mortality/severity results (Figure 3).

#### 2.1.4. Planned Intended Use

Based on the unmet medical need in the ED discussed above, Inflammatix presented the following intended use statement, which may differ from the final instructions for use (IFU):

*InSep is a gene expression profiling assay that quantifies relative expression of Host response genes from whole blood collected in the PAXgene Blood RNA tube. The test generates three scores that fall within discrete interpretation bands. These scores assess (1) the likelihood of bacterial infection, (2) the likelihood of viral infection, and (3) the likelihood of receipt of ICU-level care (including requirement for ICU transfer, mechanical ventilation, vasopressors, or renal replacement therapy (RRT)) within 7 days*.*The test is intended to be used at the time of suspicion of acute infection to assess the risk of bacterial and/or viral etiologies, and to act as an aid in guiding antimicrobial treatment decisions and in the risk assessment of disease progression in conjunction with clinical assessment and laboratory findings*.

## 3. Results

Based on brief introductions to the assay and instrument concept, advisors discussed the following topics: (1) intended use, (2) severity endpoints suitable for performance of the InSep test, (3) the patient journeys and sample flow in the ED, (4) assay result report and instrument requirements, (5) test performance requirements for clinical utility, (6) statistical considerations (impact of prevalence on predictive values and distribution of results into different interpretation bands), as well as (7) economic considerations. Case vignettes were used to provide real-world examples in support of discussion topics. Key quotes by advisors when commenting on the various topics of advisory panel meeting are listed in Table 1.

### 3.1. Intended Use

As the Intended Use is a critical component of the instructions for use defining the use case for a test, advisors were given the proposed Intended Use statement: “InSep is intended to assist in the evaluation of any patient suspected of an acute infection and to support clinician decisions regarding whether or not to treat with antibiotics, selection of other downstream diagnostics, and level of care”. Following appropriate regulatory approvals, it could be used at ED admission, during ward “sepsis alerts”, and/or when making decision regarding ICU status changes. InSep may also be shown to be useful for determining when to de-escalate antibiotic treatment and/or determine risk for readmission as a part of discharge planning. It was proposed that it might be considered as part of a daily testing for inpatients with known and/or suspected infections.

The panel confirmed that the ED was the appropriate initial setting for validation of the InSep test. The ED is the source of 50–70% of sepsis cases. Rapid and accurate identification of these severe patients as a “rule-in” is of high clinical relevance. At the same time, the ED is also the setting where less-severe patients are seen who could be rapidly “ruled out”. Providing a tool which can deliver this data will facilitate actionable decision by the ED physician, such as de-escalating level of care to an observation unit or regular ward. Advisors stated that “hospitals are scrambling due to the CMS mandates surrounding sepsis (e.g., pay for performance, bundled payment) and indicated that triaging and initial care help is needed to discriminate ‘infected vs. non-infected patients’. Furthermore, the advisory panel felt that “current alerts in electronic medical record are overly sensitive.” Advisors unanimously agreed that InSep (while ambitious) has the potential to assist with triage and make alerts more pragmatic. The advisors did raise several important additional questions, including the validity of InSep in immunocompromised patients, co-infections of bacterial and viral pathogens, atypical bacterial infections as causes of pneumonia, and changes in host-response gene expression over the time course of an infection.

Advisors were recruited from diverse specialties and hospitals, from EDs serving between 120 to 400 patients per day. The advisors agreed that, in their ED, approximately 30% of patients with suspected infections would be candidates for testing with InSep during the warmer months, whereas in the winter months, the percentage might be as high as 70% or more. The percentage would increase if febrile children were included in the intended use population. The use of molecular tests (e.g., syndromic pathogen panels) for assessment of suspected infections was established in approximately half of advisors’ institutions, most often to support the decisions on hospital admission and whether or not to send a patient to the ICU. The advisors showed an interest in the integration of InSep result into clinical decision algorithms, i.e., in combination with advanced microbiological tests such as syndromic molecular panels and metagenomic sequencing. Of interest, none of the advisors used PCT routinely in adult settings.

### 3.2. Appropriate Severity Endpoints

The advisors discussed several potential appropriate risk-stratification endpoints with regards to outcome measures and time points for use of InSep in the ED. In terms of outcome measures, 30-day mortality, need for ICU interventions, length of stay, and readmission were identified as the most valuable endpoints for management of patients in the ED. Outcome prediction timeframes from 24 h to 30 days were discussed; however, predicting near-term outcomes (e.g., 1, 2, 7 days) was noted as most relevant and important from the ED vantage with regard to influencing care, i.e., transfer to ICU and resource allocation. Advisors noted that 30-day mortality is useful to compare the performance of InSep to other existing tests but is likely less valued as a clinical useful endpoint for use in an ED setting. A composite endpoint was discussed as an option, but some advisors warned about the use of composite endpoints (“there’s always one more important than the other”). There is also exhaustive literature from other fields, i.e., cardiology, where composite endpoints were often difficult to achieve in clinical trials [37]. Outside of the ED, ICU patient monitoring and determination of step down or discharge were identified as valuable endpoints.

Clinical decision tools currently used to quantify severity risk (e.g., quick Sequential Organ Failure Assessment [qSOFA]) require a delta over 48 h. The advisors generally identified a higher priority for a test which could help determine whether a patient could be safely discharged to home, rather than in determining whether sicker patients should be admitted to the ward versus the ICU. Still, it was generally agreed that prediction of organ dysfunction and decompensation over a short period (up to 72 h) could potentially improve level-of-care decision making, for closer monitoring and/or need for immediate or downstream ICU transfer.

### 3.3. Beyond the ED, Additional Patient Populations of Greatest Need

In regard to clinical settings where InSep could be adding value outside of the ED, advisors strongly felt that the InSep test could also assist in decision making around antimicrobial stewardship in multiple patient cohorts; this would include strategies to prevent the overuse of antibiotics and to support de-escalation of treatment. In addition to patients in the ED, assessment of patients who present at urgent care centers, minute clinics and general practitioner offices within their networks were called out by the advisors as important settings for the use of InSep, given that antibiotics are highly overused in these settings [38]. Lastly, febrile children and patients with neutropenic fever were recognized as other important patient populations for use of InSep, as overuse of antibiotics is also common in these patients [39,40].

Advisors identified as additional areas of unmet medical need postoperative patients (where initiation of antibiotics is often a “guessing game”), patients frequently transferred to the hospital from long-term care facilities with nonspecific signs, e.g., altered mental status (to distinguish between infections and other etiologies), and monitoring of patients in the ICU or hospital ward.

### 3.4. Patient Journeys and Sample Flows in the ED and Beyond

EDs face significant challenges in delivering high-quality and timely patient care on a background of increasing patient numbers and limited hospital resources (“unique operation, optimized to exist at the edge of chaos”). The use of physician triage, rapid assessment, and streaming have been shown to improve patient flow [41]. In addition, when used effectively, point-of-care testing has been shown to reduce patient times in the ED [41]. To better understand the location and timing of diagnostic testing in the ED, we asked advisors about the patient journey and sample flow in the ED and the laboratory.

Advisors agreed that the InSep device may be placed in the main clinical laboratory (possibly the microbiology laboratory) in settings where a rapid turnaround time can be guaranteed; the test could ultimately be placed in (near-) ED point-of-care (POC) settings to expedite turnaround times. Satellite ED laboratories, located in or near the ED, have been increasingly implemented for ED-specific tests requiring a rapid turnaround time including glucose meters, urine pregnancy testing, blood gases and influenza testing.

There was general consensus that initially, a test such as InSep is likely to be ordered by the treating clinician in a “protocol room” but not as part of the initial (nursing) triage activities. Advisors felt that providers will have to gain familiarity and comfort with the use of InSep and its performance before it could be integrated into triage order sets. If ultimately implemented at triage, InSep would have to be inserted into the existing clinical and sample workflows and ED/ICU teams (and the clinical and administrative leadership) would have to develop new protocols through review of the data and consensus decision making. It was generally viewed that there could be high clinical value if this triage implementation were ultimately adopted.

Essential factors for success of protocol changes in the ED consist of obtaining the support from all professionals and providing ongoing education; at present, protocol changes in the management of patients with suspected sepsis remain challenging due to the lack of accurate tools to identify sepsis [42]. InSep was thus judged likely to need extensive physician education to influence patient care. However, key decisions that could be supported by InSep include antibiotic use and hospital admission, both areas where there is currently a major unmet medical need. Thus, advisors universally endorsed the InSep test to be of high clinical utility in their institution.

Patients meeting criteria for sepsis (such as based on qSOFA or Systemic Inflammatory Response Syndrome [SIRS] criteria) typically receive a specific set of tests mandated by the CMS SEP-1 Bundle in the US [20]. Advisors agreed that InSep should be used for initial sepsis assessment for the purposes of: determining whether to start antibiotic therapy or not; determining the need for other tests; and helping with decision support to providers regarding level of care, from discharge to ICU transfer. InSep could also potentially eliminate other tests (e.g., if InSep showed a viral result, no C-reactive protein (CRP) or procalcitonin (PCT) would be ordered). The advisors mentioned that electronic health records (EHR) sepsis “sniffers” often lack specificity (causing alarm fatigue and overtreatment), and that InSep might be a helpful as a reflex to an EHR-based alarm [42].

Using InSep to diagnose patients suspected of infection in the ICU or hospital ward was also mentioned. First, the test was seen as valuable in supporting decisions to discharge patients from the ICU or hospital ward; the test could show clinical utility with lingering patients in the ICU as part of the initial bundle of tests ordered given that ID physicians or hospitalists frequently are making decisions in the ICU, especially with postoperative patients (e.g., high white count, tachycardia). Here, the test could serve as an adjunctive diagnostic tool regarding presence or absence of an infection, and guide providers to appropriately prescribe antibiotics.

### 3.5. Assay and Instrument Requirements

To address the needs of future users, customer requirements gathered by Inflammatix were presented and discussed. A test turnaround time of 20–30 min was considered good. A 1-h turnaround time is becoming the standard for most molecular platforms, e.g., syndromic molecular panels, to allow adjustment of antibiotics after the first dose. A hands-on time of roughly one minute without any pipetting was considered mandatory to allow use in the ED at the point of care; outside the ED, e.g., in the near-point-of-care central or satellite laboratory a hands-on time of 5 min or less would be acceptable.

Other critical requirements identified were the need for a small instrument footprint, storage of reagents and operation of the device at room temperature, easy processing and maintenance, as well as state-of-the-art quality and IT features (including safe automated electronic transfer of information into the EHR) comparable to platforms currently used for syndromic molecular panels. The advisors stated that instrument throughput will not be an issue as the instrument would probably not be used more than 3 times per hour. However, as with other near-POC instruments, there was a willingness for onboard multiple units if throughput demands required that. Advisors were also interested in the work and training requirement associated with control of InSep, e.g., the need for daily quality controls, availability of well-characterized validation specimens from third-party vendors, and validation assistance for quantitative cutoffs near the limit of detection.

Software updates to the instrument provided by the manufacturer would be permitted at institutions if accompanied by notification and education; these updates would likely need to be executed locally (not via remote connection) given strict hospital security protocols, and changes made would have to undergo validation.

### 3.6. Result Readout and Case Vignettes

Clinical decision making is highly dependent on the ease of interpretation of laboratory tests. Advisors were therefore shown proposed patient reports for InSep (Appendix A). There was consensus that adoption of a new test in the ED setting will only be successful if the result readout is easy to understand and actionable. However, only a simplified patient report (combination of numbers and letters) would be suitable for the EHR. Therefore, providing results in words (e.g., likely, unlikely, indeterminate) would be a preferred simplification.

To demonstrate how patient reports may be useful in the management of ED patients, advisors were shown a patient case vignette as follows:

A 69-year-old female patient presents to the ED with a complaint of subjective fever. Her medical history is significant for diabetes mellitus, and she is an active smoker. Her vitals on presentation are: pulse 95, blood pressure 125/83, respiratory rate 25, temperature 37.5 °C.

For this patient presentation, bacteria and viruses can be considered as causative pathogens, and the severity could range from low to high requiring distinct management decisions. Three distinct result combinations of InSep were envisioned for the same case presentation (Figure 4A–C).

The advisors voiced approval for the potential utility of being shown the three scores at once. They further discussed the value of presenting assay results in different interpretation bands, ranging from very unlikely to very likely, for each readout (bacterial, viral, and severity) based on likelihood ratios and numerical scores. One advisor opined that the result report may help to change provider behavior since, in his opinion, approximately 75% of physicians would prescribe an antibiotic to a patient with fever and a low risk of viral infection; while even in patients with a positive influenza test, 50% of patients today would still be prescribed antibiotics. Scores were felt to potentially be useful for longitudinal tracking of patients’ status.

### 3.7. Test Performance Requirements for Clinical Utility

After introduction to InSep background, test concept, and potential result readouts, advisors were questioned about the desired performance characteristics of the test. In order to assess accuracy requirements for clinically actionable test results and successful adoption in the ED, the presentation for the advisors included a review of the basics of interdependencies and benefits of individual statistical tests, including sensitivity, specificity, likelihood ratios. Additionally, examples of the impact that prevalence of infection has on negative and positive predictive values were presented to the advisors, prior to discussion of how the InSep test results could best be presented to clinician users and what the requirements for InSep performance characteristics are.

The basic measures of the validity of binary diagnostic tests are:(a)sensitivity, which denotes the probability of a positive test in the presence of the disease of interest; and(b)specificity, which denotes the probability of a negative test in the absence of disease [43,44].

Using these measures, likelihood ratios (LRs) can be calculated where the positive LR is the probability of a positive test result in the presence of the disease divided by the probability of a positive test result in the absence of the disease [45]. Analogously, the negative LR is the probability of a negative test result in the presence of the disease divided by the probability of a negative test result in the absence of the disease.

When taking prevalence (pre-test probability of disease) into account, measures of particular interest in clinical practice are predictive values which denote post-test probability of disease given a certain test result [46]. Positive predictive value (PPV) is the probability of presence of disease given a positive test result, and negative predictive value (NPV) is the probability of absence of disease given a negative test result. Lastly, the plot of sensitivity versus 1-specificity is called the receiver operating characteristic (ROC) curve and the area under the ROC curve (AUROC) is often used as an overall measure of a diagnostic test’s accuracy [47] without the need for cutoffs.

Overall, mathematical interdependencies of sensitivity, specificity, NPV, PPV and LRs are complex, and providers without detailed statistical training often find them difficult to interpret. One issue is understanding how test performance characteristics must be combined with pre-test probability when deciding how to manage a particular patient given that their test result is critical. These complexities underscore the need for thorough education of end users and simple clear messaging for the InSep test performance characteristics overall but most important for the specific result readouts and their clinical utility.

In addition to the performance characteristics, the advisors were informed that the percentage of overall InSep test results falling into the two outer bands labeled very unlikely (low or rule-out band) and very likely (high or rule-in band) (see Figure 3 and Figure 4 above and Table 1 below) should be maximized for healthcare professionals in order for them to perceive the test as having added value, relevant for driving practice adoption. However, it was also noted that while a test cutoff can be set to make a result more stringent, it comes at the expense of putting fewer patients in the stringent band (e.g., one could set a PCT positive threshold to 50 ng/mL, ensuring a very high specificity for the rule-in band, but only a very small number of patients would ever be ruled in).

Overall, to attain the full value of the test, Inflammatix decided that defining the thresholds for the individual bands based on LRs is of paramount importance, especially for the very unlikely and unlikely bands that would allow ruling out a bacterial infection. The test’s performance characteristics must then be compared to that of other biomarkers such as PCT to determine benefit over existing tools.

To obtain information on required performance characteristics of InSep and tolerance levels to harms and unintended costs of incorrect results, advisors were presented with tables to assess the utility of varying cutoffs for individual bands as well as varying prevalences of infection. Lessons learned are only shown for the bacterial infection score but can be extrapolated to the other InSep result readouts (viral infection score, severity score).

#### 3.7.1. Impact of Prevalence on Predictive Values and Distribution of Results into Different Interpretation Bands

The advisors were provided with data reviewing the fact that variations in patient populations as well as seasonal infection rates affect the prevalence of bacterial and viral infections, which, in turn, affect the performance characteristics of all microbiological tests. Table 2 shows modelled InSep performance characteristics including sensitivity, specificity, and how a change in prevalence of bacterial infection changes the NPV and PPV of the test as well as the proportion of patients falling into each interpretation band. Of interest, at a prevalence of 50% (chosen arbitrarily to drive the discussion but representative for certain settings), the PPV for the very likely interpretation band is 91%, indicating that less than 1/10 of patients are erroneously classified as very likely bacterial infection); in contrast, at a prevalence of only 20%, the PPV for the very likely interpretation band is 71%, indicating that nearly 1/3 of patients are erroneously classified as very likely bacterial infection (but 3.5 times higher than pretest probability). The NPV for the very unlikely interpretation band is 95% versus 99%, respectively. The proportion of patients falling into the very likely and very unlikely interpretation bands also changes from 50% at a 50% prevalence to 44% at a 20% prevalence. These example calculations demonstrate the marked impact of prevalence on multiple test parameters, i.e., predictive values and the proportion of patients falling into different interpretation bands.

Advisors felt that a prevalence of 50% for bacterial infections was extremely high, and in that setting, InSep may not be useful to change clinical actions outside of the very unlikely band, i.e., most physicians would initiate antibiotic therapy for most patients. However, in lower-prevalence settings which are more common, the different bands were viewed as useful.

#### 3.7.2. Impact of Varying Interpretation Band Cutoffs and AUCs on Performance Characteristics and Distribution of Results into Different Interpretation Bands

Whereas the prevalence impacts test performance and proportion of patients falling into individual interpretation bands independent of the test design, the selection of LR cutoffs for the different interpretation bands is a major parameter affecting performance which can be changed by test design. More ambitious choices of LRs for the outermost informative bands (very likely and very unlikely) improve their post-test predictive values, but at the cost of more patients falling into the less-informative middle bands instead of the highly informative outer bands. In this regard, Table 3 shows example calculations for test results at a fixed prevalence of 50% with varying LRs for the very unlikely (rule-out) interpretation band.

Changing the cutoff for the very unlikely interpretation band to lower the LR from 0.15 to 0.05 markedly improves its sensitivity (98% vs. 89%) and NPV (95% vs. 87%). As a trade-off, a more rigid cutoff of 0.05 results in only 19% of patients falling into the very unlikely interpretation band, compared to 44% if using a cutoff of 0.15.

Appendix A show similar calculations for InSep results at a fixed prevalence of 50% and 20% with varying likelihood ratios for the very likely (rule-in) interpretation band. When the very likely (rule-in) interpretation band LRs are lowered from 10 to 5, the specificity drops from 94% to 83%, resulting in PPVs of 91% versus 83%, respectively. Again, the proportion of patients in the very likely band increases from 31% at an LR of 10 versus 50% at an LR of 5.

The overall discriminatory ability of the diagnostic score leads to the degree of trade-off seen above in stringency versus number of patients in band. Table 4 shows results for three hypothetical InSep bacterial score AUCs of 0.9, 0.87, and 0.85, each used in a setting with 50% infection prevalence and each with the same target LRs for the very unlikely and very likely interpretation bands of 0.075 and 10, respectively. With an AUC of 0.9, the proportion of patients falling into these interpretation bands is >60%. In contrast, at an AUC of only 0.85, the high LR of 10 is impossible to reach. Furthermore, the proportion of patients falling into the outer interpretation bands is only 18%. To achieve the same proportion of patients in the very unlikely and very likely interpretation bands as the InSep test with an AUC of 0.9, InSep with an 0.85 AUC would require adjustments to the cutoffs for the very unlikely and very likely bands that would make these bands significantly less informative.

When discussing the examples shown above, some advisors challenged the concept of four interpretation bands and felt that three bands (unlikely, indeterminate, and likely) would be easier to understand. However, given diagnostic odds of 5–10 between the middle two bands, it was agreed that some stratification was present, at the trade-off of complexity of interpretation. Advisors noted that a test report should be simplified and provide results in words (e.g., likely, unlikely, indeterminate) rather than numbers, which was felt to be more compatible with EHR and increase the likelihood that results with be acted upon.

Advisors felt most comfortable in using sensitivity and specificity, given that NPV and PPV are dependent on disease prevalence and thus are not useful to the average clinician who is unaware of prevalence. In the context of clinical practice, advisors emphasized that high sensitivity is most critical to minimize incorrect calls on septic patients; they felt a strong need to guard against a scenario in which no antibiotics are recommended but the patient may actually need to be treated. They generally viewed negative LRs of between 0.05 to 0.075 for the very unlikely band as stringent enough to form a strong rule-out result that could change clinical practice. At the same time, advisors would like to see high specificity for the rule-in bands. Positive LRs of between 5 and 7.5 would drive clinical actions, i.e., ruling-in for antibiotics. Also, there was general consensus that, at a test specificity of 86%, a result in the likely band would prompt treatment with an antibiotic. Interestingly, advisors agreed that looking at only one of the three InSep result readouts leaves 2/3 of the information InSep provides untouched; the inclusion of results for severity and the risk for viral infections in InSep could further influence clinical decision making (i.e., discussing test characteristics of the bacterial score alone did not allow for full contextualization).

## 4. Discussion

The clinical advisory panel identified a clear demand for a diagnostic solution such as the InSep test. The advisory panel pointed out several key unmet medical needs based on the current equipoise when seeing patients with suspected infections and/or suspected sepsis, especially in the ED [5,11].

First, the panelists agreed that as current diagnostic offerings are insufficient for the rapid diagnosis of acute infections and sepsis, the InSep test with its three independent readouts for the likelihoods of bacterial and viral infections and severity of the condition would provide actionable results that have the potential to guide decision making. Panelists agreed that the presented levels of performance would be clinically actionable for ruling in and ruling out infections, in particular decision making on whether to prescribe antibiotics. The performance parameters targeted by the InSep test were believed to be appropriate boundaries for assisting clinicians with antimicrobial stewardship efforts relevant for decreasing emergence of antimicrobial resistance [48], while also identifying those patients who require appropriate antibiotics as mandated by the CMS sepsis bundle [20]. The panel also believed that addition of the viral score could help with infection containment (particularly during influenza season or a viral pandemic), while the severity score was viewed as a useful adjunct in level-of-care decisions, if presented with an appropriate short-term readout.

Second, advisors were excited by the possibility that InSep may identify patients with noninfectious inflammation, or with bacterial–viral co-infections who would have been missed previously, such as patients with bacterial superinfections on top of severe viral infections.

Third, the panel noted that the diagnostic readouts of the InSep test may have a marked impact on the use of other differential diagnostic tests. If used early in the patient journey (e.g., at triage), the rule-in and rule-out results for bacterial and viral infections could change the course of actions taken by the treating clinician. A “very likely” InSep result for bacterial or viral infection could result in early escalation by rapidly ordering a pathogen detection test, such as a rapid flu test or a syndromic respiratory panel test. Conversely, the InSep “very unlikely” result may help to avoid unnecessary test orders for infections (e.g., syndromic panels or biomarkers like PCT). At present, the lack of a readout for fungal infections was not identified as a major limitation as fungal infections are less frequent in a typical ED population.

Fourth, the unique severity readout of the InSep test may assist in level-of-care decisions. As the currently available tests (i.e., lactate) and clinical scores are sensitive but not highly specific, many patients are unnecessarily admitted. The InSep test could rule out severe conditions and allow to safely discharge the patient, thereby assisting the attending in the ED and saving costs to the healthcare system. At the same time, the test could help identify those patients at high risk for sepsis not otherwise suspected of the condition. Earlier and more accurate identification of these patients should allow escalation of care and improve outcomes.

The InSep test is currently being developed as a rapid point-of-care/Clinical Laboratory Improvement Amendments (CLIA)-waivable platform with a turnaround time of less than 30 min. The advisors strongly felt that these characteristics were highly useful to enable the rapid diagnosis of acute infections and sepsis as well as the prognosis of the condition’s severity. The advisors pointed out that rapid interpretability of the result as a directly actionable call would be needed to successfully introduce the InSep tests into clinical algorithms. This implementation would require careful planning and a major effort in medical education due to the innovative but complex test result readout. The test may therefore be introduced initially into the central laboratory rather than the ED or a ward/ICU. Based on the intended use presented, the advisors agreed that the patient populations of greatest need identified was the patient in the ED; advisors also indicated that ICUs [49] and long-term care facilities [50] would be settings that could benefit from the value InSep brings, combining diagnostic and prognostic readouts in a single test.

Supporting the findings of the advisory panel presented above, a web-based survey among US ED physicians about current sepsis diagnostic practices revealed that 100% of respondents use complete blood counts, lactate, and blood cultures, between 95% and 99% use comprehensive metabolic panels, chest X-rays, and urine cultures; in contrast, PCT and CRP were only used by 20% of respondents [51]. The severity of suspected sepsis is commonly assessed using SIRS criteria (77%), whereas the SEP-1-bundle measures are rarely applied (14%). A high rate of dissatisfaction with current diagnostic tools for sepsis and demand for improved diagnostics was expressed. When presented with the blinded product profile for the InSep test, the vast majority found the test’s performance clinically robust and of clinical utility and would likely recommend it for incorporation into their respective institution’s sepsis protocols.

In regards to the health-economical value of the InSep test, we have recently reported substantial savings expected with the introduction of InSep among adult patients with acute respiratory tract infections in EDs in the US [52]. A cost impact model estimated costs for these patients versus with the adoption of InSep was superior to standard of care (including procalcitonin), resulting in average cost savings of nearly $2000 per patient tested, exclusive of the cost of the InSep test. Reductions in hospital days, antibiotic days and 30-day mortality were driven by the InSep test providing fewer “noninformative” moderate risk predictions and more “certain” low- or high-risk predictions. Further studies, including interventional studies, are planned to confirm these results.

Of note, Inflammatix recently used the identical approach of mining large data sets containing diverse cohorts of patients with viral illnesses including influenza to develop a 6-RNA-host response signature for the severity of viral illnesses from whole blood; the signature was successfully shown to predict the severity of COVID-19 and may allow improved decision making regarding patient dispositions (safe discharge or escalation) [53]. While our advisory panel was conducted prior to COVID-19, the implementation of InSep at triage for suspected COVID-19 would likely aid in quickly identifying which patients need quarantine, which have bacterial or noninfectious inflammation, and how severe the condition is. Host response research, when advanced to commercially available diagnostic tests, may favorably impact healthcare resource utilization and improve patient care during COVID-19 and emerging pandemics.

The results of the advisory panel presented here have limitations. First, the information presented to the advisors was in some cases preliminary and/or hypothetical (i.e., performance characteristics and instrument design and specifics), thus introducing potential bias. Second, the selection of advisors for the advisory panel meeting may not have been representative of the intended user population, as advisors had diverse backgrounds ranging from emergency medicine to infectious diseases, intensive care, and laboratory medicine. Also, all participants were practicing in North America and all but one came from academic institutions, therefore potentially underrepresenting the needs of community and other less-specialized medical settings.

## 5. Conclusions

The advisors identified a clear need for improved diagnostic offerings for acute infections and sepsis and provided insight into how the InSep test and other novel diagnostics can provide the most clinical utility in this challenging patient population. Further physician and market research will be incorporated into the final device to ensure that the InSep test fits into rapid workflows and provides accurate information. Rigorous prospective multicenter clinical studies are currently being planned and conducted to demonstrate that the theoretical performance presented here remains true across broad populations. 

## Figures and Tables

**Figure 1 jpm-10-00266-f001:**
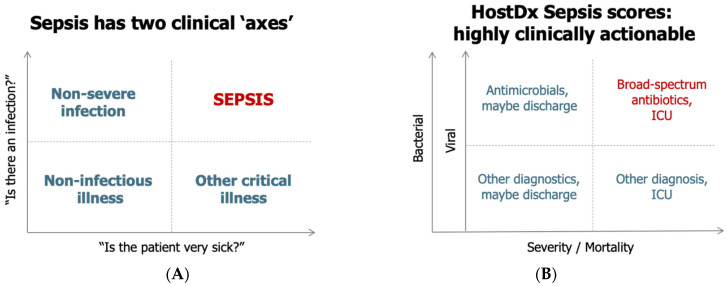
The two axes of sepsis: presence of infection and severity (**A**), and the broad and distinct clinical actions associated with the two axes (**B**). ICU: intensive care unit.

**Figure 2 jpm-10-00266-f002:**
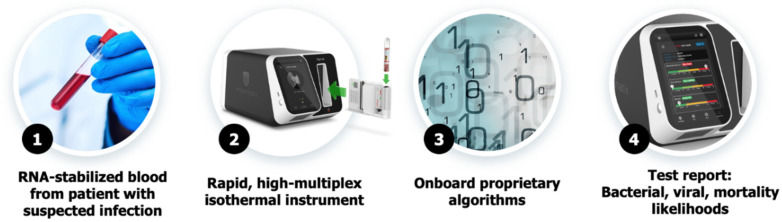
Workflow as presented for the InSep test from sample collection via PAXgene Blood RNA vacutainer (**1**) to loading of the cartridge into the instrument (**2**), processing on the instrument including the proprietary algorithm (**3**) and result reporting (**4**).

**Figure 3 jpm-10-00266-f003:**
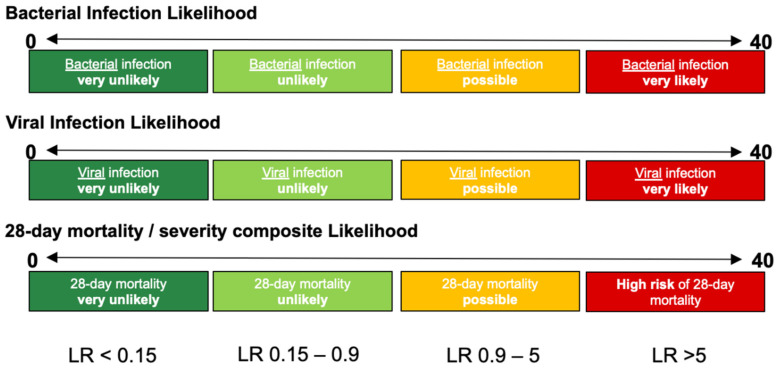
Proposed InSep result readout: Three separate results are provided, each containing an absolute score falling into one of the interpretation bands. There are 4 bands each for the bacterial, viral, and the severity score. Each band has different target likelihoods of the corresponding event.

**Figure 4 jpm-10-00266-f004:**
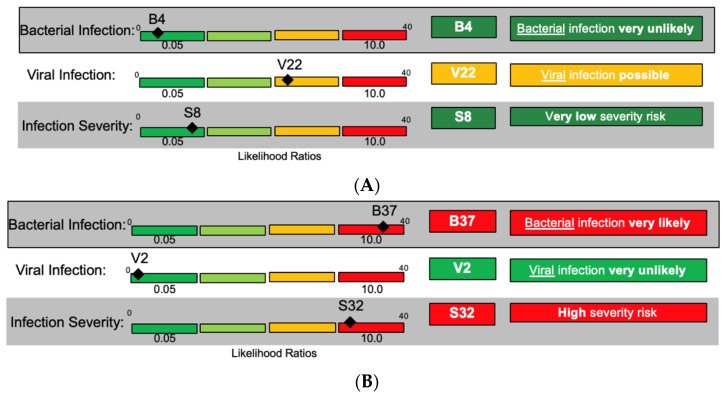
Three hypothetical result readouts (**A**–**C**) for InSep for the same patient presenting with subjective fever and vital sign changes.

**Table 1 jpm-10-00266-t001:** Key quotes from advisory panel members.

Topic	Quote *
Center for Medicare Services (CMS) mandates/bundles	*“Hospitals are scrambling due to the Centers for Medicare & Medicaid Services (CMS) mandates surrounding sepsis (e.g., pay for performance, bundled payment); therefore, help is needed for triage to discriminate infected vs. non-infected patients.”*
EMR sepsis alerts	*“Current alerts in electronic medical record are overly sensitive.”*
Diagnostic use cases for InSep	*“The ED is the appropriate initial setting for validation of the InSep test.”*
	*“InSep has the potential to assist with triage and make alerts more pragmatic.”*
	*“The decision to continue/stop antibiotics could benefit from use of InSep.”*
	*“30% of patients with suspected infections would be candidates for testing with InSep during the warmer months whereas in the winter months the percentage might be as high as 70% or more.”*
	*“75% of physicians would prescribe an antibiotic to a patient with fever and low risk of viral infection. Currently, with a positive flu test, 50% of patients are still prescribed antibiotics.”*
	*“I* *nformation providing a high level of confidence that a patient did not have a bacterial infection would eliminate unnecessary antibiotic treatment.”*
	*“There is interest in the integration of InSep result in clinical decision algorithms,* i.e., *in combination with advanced microbiological tests such as syndromic molecular panels and metagenomic sequencing.”*
Prognostic use cases for InSep	*“A tool that accurately predicted which patients would not require transfer to the intensive care unit (ICU) was very important.”*
	*“There is a greater need in determining whether to discharge a less severe patient rather than in determining whether sicker patients should be admitted to the ward versus the ICU.”*
Value of other biomarkers	*“We do not feel very strongly about the use of procalcitonin, other than in pediatrics.”*
	*“If InSep showed a viral result no C-reactive protein (CRP) or procalcitonin (PCT) would be ordered.”*
Value of syndromic molecular panels	*“We often use syndromic pathogen panels for assessment of suspected infections to support decisions on hospital admission and whether or not to send a patient to the ICU.”*
Endpoints of importance to ED doctors	*“30-day mortality, need for ICU interventions, length of stay, and readmission are the most valuable endpoints for management of patients in the ED.”*
	*“Prediction of organ dysfunction and decompensation over a short period (up to 72 h) could potentially prompt closer monitoring and possible ICU transfer.”*
	*“Predicting near term outcomes between 1, 2 and 7 days are important to compare performance of HostDx Sepsis to other tests.”*
	*“I would act upon the negative LR of 0.05 for a ‘very unlikely’ result, I want to be confident in ruling patients out.”*
Result presentation and accuracy requirements	*“I want high specificity in cases where bacterial infection is likely (e.g., intravenous drug user), and high sensitivity in a patient unlikely to have bacterial infection (e.g., a patient presenting from a long-term care facility). Sensitivity and specificity are each important and their relative importance varies with patient characteristic.”*
	*“Negative LRs of between 0.05 to 0.075 for the very unlikely band are appreciated as they would make the very unlikely band a rule-out results and would thus allow changing clinical practice.”*
	*“It is important to guard against a scenario in which you recommend no antibiotics, but the patient needed them.”*
	*“I use sensitivity and specificity, as NPV and PPV aren’t useful if I do not have baseline values against which to compare test results.”*
Instrument requirements	*“A turnaround time of 30 min would be fine … a decision can wait until after first antibiotic dose if necessary.”*
	*“There is a need for characterized validation specimens from a third-party vendor. Providing validation assistance and quantitative cut-offs is important, ensuring they are near the limit of detection is important.”*
Overall value of InSep	*“Getting InSep into ED at all is a win.”*
	*“Inclusion of results for severity and the risk for viral infections in InSep could influence clinical decision making.”*

*, We use here the test name InSep. Advisors were not introduced to the rebranded test name, InSep, but the former name HostDx Sepsis during the advisory panel meeting.

**Table 2 jpm-10-00266-t002:** Impact of bacterial infection prevalences of 50% (**A**) and 20% (**B**) on the proportion of patients falling into InSep interpretation bands and positive and negative predictive values based on modelled data. We applied a negative likelihood ratio (LR) cutoff of 0.05 and a positive LR cutoff of 10, assuming these were clinically actionable.

(A)								
Prevalence = 50%, LR low = 0.05, LR high = 10								
	Neg	Pos	LR	% in Band	NPV	PPV	Sens	Spec
Very likely	28	281	10.0	31%	68%	91%	56%	94%
Possible	73	151	2.07	22%	55%	68%	30%	86%
Unlikely	218	59	0.27	28%	79%	61%	88%	44%
Very unlikely	181	9	0.05	19%	95%	61%	98%	36%
**(B)**								
**Prevalence = 20%, LR low = 0.05, LR high = 10**								
	**Neg**	**Pos**	**LR**	**% in Band**	**NPV**	**PPV**	**Sens**	**Spec**
Very likely	45	112	10.0	16%	90%	71%	56%	94%
Possible	116	60	2.07	18%	83%	34%	30%	86%
Unlikely	349	24	0.27	37%	94%	28%	88%	44%
Very unlikely	290	4	0.05	29%	99%	28%	98%	36%

**Table 3 jpm-10-00266-t003:** Calculations for InSep results at a fixed prevalence of 50% but varying likelihood ratios for the low (very unlikely = rule-out) interpretation band ranging from 0.05 (**A**), 0.075 (**B**) to 0.1 (**C**) and 0.15 (**D**) based on modelled data.

(A)								
Prevalence = 50%, LR low = 0.05, LR high = 10								
	Neg	Pos	LR	% in Band	NPV	PPV	Sens	Spec
Very likely	28	281	10.0	31%	68%	91%	56%	94%
Possible	73	151	2.07	22%	55%	68%	30%	86%
Unlikely	218	59	0.27	28%	79%	61%	88%	44%
Very unlikely	181	9	0.05	19%	95%	61%	98%	36%
**(B)**								
**Prevalence = 50%, LR low = 0.075, LR high = 10**								
	**Neg**	**Pos**	**LR**	**% in Band**	**NPV**	**PPV**	**Sens**	**Spec**
Very likely	28	281	10.0	31%	68%	91%	56%	94%
Possible	79	155	1.97	23%	55%	66%	31%	84%
Unlikely	118	43	0.36	16%	73%	55%	91%	24%
Very unlikely	275	21	0.07	30%	93%	68%	96%	55%
**(C)**								
**Prevalence = 50%, LR low = 0.10, LR high = 10**								
	**Neg**	**Pos**	**LR**	**% in Band**	**NPV**	**PPV**	**Sens**	**Spec**
Very likely	28	281	10.0	31%	68%	91%	56%	94%
Possible	72	150	2.08	22%	55%	68%	30%	86%
Unlikely	72	36	0.49	11%	67%	52%	93%	14%
Very unlikely	328	33	0.1	36%	91%	73%	93%	66%
**(D)**								
**Prevalence = 50%, LR low = 0.15, LR high = 10**								
	**Neg**	**Pos**	**LR**	**% in Band**	**NPV**	**PPV**	**Sens**	**Spec**
Very likely	28	281	10.0	31%	68%	91%	56%	94%
Possible	44	119	2.73	16%	55%	73%	24%	91%
Unlikely	44	42	0.95	9%	51%	50%	92%	9%
Very unlikely	385	58	0.15	44%	87%	79%	89%	77%

**Table 4 jpm-10-00266-t004:** Impact of varying AUCs on test characteristics using modelled data. Shown are estimated performance at AUCs of (**A**) 0.9, (**B**) 0.87, and (**C**) 0.85 at a prevalence of 50%. Here, we had LRs fixed at 0.075 for the very unlikely band and at ~10 (or as close as possible) for the very likely band.

(A)								
AUC: 0.90								
	Neg	Pos	LR	% in Band	NPV	PPV	Sens	Spec
Very likely	28	281	10.0	31%	68%	91%	56%	94%
Possible	98	168	1.7	27%	55%	63%	34%	80%
Unlikely	98	31	0.31	13%	76%	54%	94%	20%
Very unlikely	275	21	0.07	30%	93%	68%	96%	55%
**(B)**								
**AUC: 0.87**								
	**Neg**	**Pos**	**LR**	**% in Band**	**NPV**	**PPV**	**Sens**	**Spec**
Very likely	11	108	10.0	12%	56%	91%	22%	98%
Possible	126	324	2.57	45%	68%	72%	65%	75%
Unlikely	189	55	0.29	24%	78%	59%	89%	38%
Very unlikely	174	13	0.07	19%	93%	60%	97%	35%
**(C)**								
**AUC: 0.87**								
	**Neg**	**Pos**	**LR**	**% in Band**	**NPV**	**PPV**	**Sens**	**Spec**
Very likely	11	108	10.0	12%	56%	91%	22%	98%
Possible	126	324	2.57	45%	68%	72%	65%	75%
Unlikely	189	55	0.29	24%	78%	59%	89%	38%
Very unlikely	174	13	0.07	19%	93%	60%	97%	35%

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
