# Peer review of "A Multi-mRNA Host-Response Molecular Blood Test for the Diagnosis and Prognosis of Acute Infections and Sepsis: Proceedings from a Clinical Advisory Panel"

_jpm, 2020, doi:10.3390/jpm10040266_

Round 1
Reviewer 1 Report
The article 'A multi-mRNA host-response molecular blood test for the diagnosis and prognosis of acute infections and sepsis: proceedings from a clinical advisory panel' presented that a 29-host-mRNA blood-based diagnostic, InSep™ that combines machine learning algorithms with a rapid point-of-care enabled platform and a turnaround time of less than 30 minutes. It is unclear how to separate leukocytes, how to evaluate mRNA in leukocytes, and what kind of molecule is checked as 29-host-selected mRNA. It would be well to be able to show how mRNA is properly evaluated by preventing contamination with RNase in emergency situation. In addition, it had better show 29 mRNAs selected and a basis for whether the selection is appropriate.
Author Response
We appreciate the reviewer's valuable comments. We believe that the reviewer’s requests are partly triggered by the “unusual” format of the manuscript representing the proceedings of an advisory panel. However, we revised the research design, the description of methods, and revised the presentation of results to facilitate the understanding of the proceedings format and overall findings. We also added information to the methods and results sections regarding the format of the InSep test including the choice of collection tubes (PAXgene Blood RNA) and the selection of target genes. We also rephrased the conclusions to be better support by results.
Specific comments:
The article 'A multi-mRNA host-response molecular blood test for the diagnosis and prognosis of acute infections and sepsis: proceedings from a clinical advisory panel' presented that a 29-host-mRNA blood-based diagnostic, InSep™ that combines machine learning algorithms with a rapid point-of-care enabled platform and a turnaround time of less than 30 minutes. It is unclear how to separate leukocytes,
In section 2.1.1 we mention that InSep will be run from 2.5 ml of blood collected in a PAXgene RNA Blood vacutainer. This collection tubes stabilizes blood for expanded period (up to years) and is commercially available (FDA-cleared and CE-marked). For clarification, we added this information to Fig. 2 (proposed workflow) in the same section of the revised manuscript. We also added, “No leukocyte separation is carried out; RNA is extracted direct from the sample using magnetic beads in the cartridge.”
…how to evaluate mRNA in leukocytes,
As mentioned above InSep will be run from the PAXgene RNA Blood collection tube. This tube contains PAXgene solution that stabilizes RNA from whole blood after lysis of cells. There is no isolation of leukocytes performed to process mRNA in the InSep test.
…and what kind of molecule is checked as 29-host-selected mRNA.
In section 2.1.2 of our manuscript we provide references for the original publications that describe the 3 gene signatures that were combined in the InSep test. The manuscript now reads, “Each of these three previously-described scores is calculated as the difference in geometric means of the expression values of two gene ‘modules’ (one composed of over-expressed genes and the other composed of under-expressed genes). The modules are: (1) infection-up: CEACAM1, ZDHHC19, C9orf95, GNA15, BATF, C3AR1; (2) infection-down: KIAA1370, TGFBI, MTCH1, RPGRIP1, HLA-DPB1; (3) bacterial-viral-up: HK3, TNIP1, GPAA1, CTSB; (4) bacterial-viral-down: IFI27, JUP, LAX1; (5) mortality-up: DEFA4, CD163, RGS1, PER1, HIF1A, SEPP1, C11orf74, CIT; and (6) mortality-down: LY86, TST, KCNJ2. Housekeeping gene(s) act as internal controls. The above genes may change in the final product as they have not been optimized for isothermal amplification.”
…It would be well to be able to show how mRNA is properly evaluated by preventing contamination with RNase in emergency situation.
As we use the PAXgene Blood RNA collection vacutainer that is FDA-cleared and CE-marked the manufacturer has ensured that mRNA is properly stabilized and RNase is eliminated and barred from degradation of RNA.
…In addition, it had better show 29 mRNAs selected and a basis for whether the selection is appropriate.
As mentioned above the original InSep design incorporated 29 host mRNAs that were referenced in the three seminal manuscripts (listed above). The identity of the genes has been added above.
Reviewer 2 Report
Thank you for giving me the chance to review this article.
As a clinician working in ED, if future study trend must be precision medicine, the subject of this study is very important. In the field, like panels commented in this study, practical laboratory factors or indicators in patients’ diagnosis or prognosis are limited -those are procalcitonin, lactate, and SOFA scores- and somewhat inaccurate. But I am not sure and do not know exactly this kind of form is acceptable or not. This study described so many pages about 40, and I think the format also is not familiar. In other words, it is nearly pannel discussion and a report of the results by the developer. Thus, I would like to withold the decision on whether or not this paper is acceptable or not. I agree with the proposal for future further study as mentioned.
Sincerely,
Author Response
We appreciate the reviewer's valuable comments. We believe that the “unusual” format of the manuscript triggered some of the comments by reviewer 2. However, we revised the presentation of results to facilitate the understanding of the proceedings format and overall findings. We also rephrased the conclusions to be better support by results.
Specific comments:
As a clinician working in ED, if future study trend must be precision medicine, the subject of this study is very important. In the field, like panels commented in this study, practical laboratory factors or indicators in patients’ diagnosis or prognosis are limited -those are procalcitonin, lactate, and SOFA scores- and somewhat inaccurate. But I am not sure and do not know exactly this kind of form is acceptable or not. This study described so many pages about 40, and I think the format also is not familiar. In other words, it is nearly pannel discussion and a report of the results by the developer. Thus, I would like to withold the decision on whether or not this paper is acceptable or not. I agree with the proposal for future further study as mentioned.
“We thank the reviewer for the very supportive comments. We realize that the article format is unusual but wish to share the comments of the panel and description of the test with the scientific community as further context for how the test is developed. Thank you for your support.“
Round 2
Reviewer 2 Report
As I already mentioned, this draft does not look like original research. And so, if editorial board determine to accept this manuscript to be published, I agree with the decision. Thank you again for me having the chance to review this article.